

# Robust Independent Validation of Experiment and Theory:
## Rivet version 3

**Christian Bierlich[1,2], Andy Buckley[3]\*, Jonathan Butterworth[4],**
**Christian Holm Christensen[1], Louie Corpe[4], David Grellscheid[5],**
**Jan Fiete Grosse-Oetringhaus[6], Christian Gütschow[4], Przemyslaw Karczmarczyk[6,7],**
**Jochen Klein[6,8], Leif Lönnblad[2], Christopher Samuel Pollard[9], Peter Richardson[6,10],**
**Holger Schulz[11] and Frank Siegert[12]**

**1** Niels Bohr Institute, Copenhagen, Denmark
**2** Department of Astronomy & Theoretical Physics, Lund University, Lund, Sweden
**3** School of Physics & Astronomy, University of Glasgow, Glasgow, UK
**4** Department of Physics & Astronomy, University College London, London, UK
**5** Department of Informatics, University of Bergen, Bergen, Norway
**6** CERN, Meyrin, Switzerland
**7** Faculty of Physics, Warsaw University of Technology, Warszawa, Poland
**8** Istituto Nazionale di Fisica Nucleare (INFN), Torino, Italy
**9** DESY, Hamburg, Germany
**10** Institute for Particle Physics Phenomenology, Durham University, Durham, UK
**11** Fermilab, Batavia IL, USA
**12** TU Dresden, Dresden, Germany

⋆ andy.buckley@cern.ch

### Analysis contributors

A. Grohsjean, A. Hinzmann, A. Knutsson, A. Jueid, A. Bursche, A. Knutsson, A. Saavedra, A. Grecu, A. Long, A. Proskuryakov,
A. Savin, A. Descroix, A. Hrynevich, A. Lister, A. E. Dumitriu, A. R. Cueto Gomez, A. Grebenyuk, A. Contu, A. Hinzmann,
A. Morsch, A. Papaefstathiou, A. Kubik, A. Chen, A. Sing Pratap, A. Mehta, A. Karneyeu, A. Hortiangtham, A. Garcia-Bellido,
A. Heister, B. Bhawandeep, B. Alvarez Gonzalez, B. Mohanty, B. Cooper, B. Nachman, B. Smart, B. Maier, B. Uppal, B. Li,
B. Zhu, B. Bilki, C. Bertsche, C. Belanger-Champagne, C. Lapoire, C. Park, C. La Licata, C. Wymant, C. Herwig, C. Johnson,
C. Nattrass, C. O. Rasmussen, C. Vaillant, C. Meyer, C. B. Duncan, C. Buttar, C. Group, D. Reichelt, D. Baumgartel,
D. Mekterovic, D. Mallows, D. Voong, D. Ward, D. d'Enterria, D. Roy, D. Kar, D. Yeung, D. Volyanskyy, D. Burns, E. Yazgan,
E. Yatsenko, E. Bouvier, E. Barberis, E. Nurse, E. Clement, E. Fragiacomo, E. Berti, E. Sicking, E. Paradas, E. Meoni, E. Soldatov,
F. Cossutti, F. Fabbri, F. Riehn, F. Dias, F. Vives, F. La Ruffa, Frank Krauss, F. Schaaf, F. Blatt, G. Hesketh, G. Sieber, G. Flouris,
G. Marchiori, G. Majumder, G. Jones, G. Pivovarov, G. Safronov, G. Brona, H. Jung, H. Van Haevermaet, H. Caines, H. Hoeth,
H. Poppenborg, H. Wang, I. Bruce, I. Ross, I. Helenius, I. Park, I. Odderskov, I. Siral, I. Pozdnyakov, J. L. Cuspinera Contreras,
J. B. Singh, J. Linacre, J. Keaveney, J. Monk, J. Robinson, J. Kretzschmar, J. Fernandez, J. Llorente Merino, J. Leveque, J. Zhang,
J. Mahlstedt, J. Bellm, J. Haller, J. Bossio, J. Hollar, J. Stahlman, J. Rodriguez, J. Cantero Garcia, J. M. Grados Luyando,
J. Katzy, J. Thom, J. J. Goh, J. Hugon, K. Ocalan, K. Mishra, K. Nordstrom, K. Moudra, K. Bierwagen, K. Becker, K. Finelli,
K. Stenson, K. Joshi, K. Kovitanggoon, K. Rabbertz, K. Kulkarni, K. Lohwasser, K. Cormier, L. Sonnenschein, L. Asquith, L. Lan,
L. Massa, L. Viliani, L. Helary, L. Skinnari, L. Kaur Saini, L. Perrozzi, L. Lebolo, L. Kreczko, L. Wehrli, L. Marx, M. Maity,
M. Alyari, M. Meissner, M. Schoenherr, M. Sirendi, M. Stefaniak, M. Galanti, M. Stockton, M. Radziej, M. Seidel, M. Zinser,
M. Poghosyan, M. Bellis, M. Mondragon, M. Danninger, M. Verzetti, M. Goblirsch, M. Azarkin, M. Gouzevitch, M. Kaballo,
M. Schmelling, M. Schmitt, M. Queitsch-Maitland, M. Kawalec, M. Hance, M. Zakaria, M. Guchait, N. Tran, N. Viet Tran,
N. Moggi, O. Kepka, O. Gueta, O. Tumbarell Aranda, O. Hindrichs, P. Gunnellini, P. Katsas, P. Bueno Gomez, P. Katsas, P. Kokkas,
P. Gunnellini, P. Kirchgaesser, P. Spradlin, P. Bell, P. Newman, P. Ruzicka, P. Starovoitov, P. E. C. Markowitz, P. Skands, P. Wijeratne,
P. Gras, P. Lenzi, P. Van Mechelen, P. Maxim, R. Gupta, R. Kumar, R. Sloth Hansen, R. Demina, R. Schwienhorst, R. Field,
R. Ciesielski, R. Prabhu, R. Rougny, R. Kogler, R. Lysak, S. Dooling, S. Sacerdoti, S. Rappoccio, S. Dooling, S. Bhowmik, S. Baur,
S. Prince, S. Sen, S. Zenz, S. Epari, S. Todorova-Nova, S. AbdusSalam, S. Stone, S-S. Eiko Yu, S. Amoroso, S. Pagan Griso,
S. Richter, S. von Buddenbrock, S. Henkelmann, S. Schumann, S. P. Bieniek, S. Linn, S. Lloyd, S. Swift, S. Banerjee, S-W. Lee,
S. Bansal, S. Dittmer, S-W. Li, T. Burgess, T. M. Karbach, T. Martin, T. Neep, T. Umer, T. Dreyer, V. Oreshkin, V. Gaultney Werner,
V. Kim, V. Murzin, V. Gavrilov, V. Pleskot, W. H. Bell, W. Y. Wang, W. Barter, W. Erdmann, X. Janssen, Y. Qin, Y. Zengindemir,
Y-T. Duh, Y. Li, Y-H. Chang, Y-J. Lu, Z. Marshall, Z. Hubacek, Z. Jiang

## Abstract

First released in 2010, the RIVET library forms an important repository for analysis code, facilitating comparisons between measurements of the final state in particle collisions and theoretical calculations of those final states. We give an overview of RIVET's current design and implementation, its uptake for analysis preservation and physics results, and summarise recent developments including propagation of MC systematic-uncertainty weights, heavy-ion and *ep* physics, and systems for detector emulation. In addition, we provide a short user guide that supplements and updates the RIVET user manual.



## Contents

# 1  Overview

Experiments at particle colliders provide many measurements of the final state in particle collisions. These measurements range from relatively simple counts of final state particles, to cross-sections for the production of complicated final states multiply-differential in the kinematics of more complex objects such as hadronic event shapes or missing energy. These measurements are typically made in so-called "fiducial" regions — that is, within a region of phase space defined by kinematic cuts to reflect regions in which the particle detectors have high acceptance and efficiency, thus minimising model dependence, since large theory-based extrapolation into unobserved regions is not required.

Relatively small "unfolding" corrections are then often applied to account for residual instrumental effects to within some evaluated uncertainty, meaning that the results can be compared directly to particle-level predictions from Monte Carlo event generators. Unfolding is performed at the distribution rather than event level, by constructing "physics objects" such as jets from physical particles in the final state of the MC events and from there to differential observables. Our picture of what is a physical particle suitable for definition of a fiducial unfolding target is nowadays usually limited to quasi-classical colour-singlets, such as leptons direct from the hard scattering, or hadrons (and their decay descendants) formed after the fundamental quantum dynamics have lost coherence via non-perturbative effects. Alternatively, a "folding" approach can be taken, in which the efficiency and resolution of the measuring equipment are estimated within the measured phase space, and applied to particle-level predictions to allow model-to-data comparisons.

Such measurements can contain a wealth of information about the short-distance physics of the collision, as well as about the intervening soft processes such as hadronisation and underlying event. Modern theoretical calculations, within and beyond the Standard Model, allow predictions to be made which can be confronted with these measurements on a like-for-like basis. RIVET exists to facilitate such comparisons, and the physics conclusions to which they lead, by providing a set of tools to compute physical fiducial physics objects with robust and standard definitions, and an extensive library of analysis routines based on such definitions and immediately comparable to published data.

This document is intended to supplement and supersede the first RIVET user manual [1], as well as providing an overview of RIVET usage to date and a summary of recently added features in RIVET versions up to and including version 3.0. We first review the applications to which RIVET has been applied, then in Section 2 review the structure to which RIVET has evolved in its decade-long existence. In Section 3 we cover the set of major new features and functionalities since the original paper, including the cuts system, automatic use of event weight vectors and event groups, new mechanisms for full-accuracy run merging, tools for heavy-ion and *ep* physics, and tools for preservation of search analyses such as detector emulation. We conclude in Section 4 with a brief user guide intended to introduce a new user to the basics of running and writing analysis routines with RIVET.

## 1.1 Applications of RIVET

RIVET has been widely used in the development [2–8], validation [9–16] and tuning [17–19] of event generators for Standard Model processes, as well as in the study of parton density functions (PDFs) [20, 21]. Tuning generally makes use of the related Professor [22] package.

RIVET has also been used by the LHC experiments as a part of their analysis and interpretation toolkit (see, for example [23–27]), and in studies for future experiments [28–31]. It has been used for development of new analysis techniques including machine learning applications, jet substructure, boosted-particle tagging and pile-up suppression [32–35]. Extraction of SM parameters for example using TopFitter [36, 37] and other phenomenological studies of the SM [38–42] have used RIVET, and it has also been employed in searching for and constraining BSM physics [43–46], sometimes making use of the related CONTUR package [47].

The above list of references is incomplete, but serves to illustrate the wide applicability of, and demand for, RIVET functionality.

## 2 Structure and design

RIVET is structured in a layered fashion, with a C++ shared library at its core, supplemented by C++ "plugin" libraries containing collider analysis routines, a Python programming interface built via the Cython system, and finally a set of Python and shell scripts to provide a command-line interface. The principle deployment targets are Unix-like systems, primarily Linux and Mac OS. RIVET's design is motivated by ease of use, in particular aiming to provide a natural & expressive analysis-writing interface with minimal technical "boilerplate", as far as possible while also being computationally efficient and supporting a wide range of use-cases.

### Dependencies

The core library provides machinery for structuring "runs" of the code, i.e. the feeding of simulated collider events into it for analysis, and for output of histogram data. These input and output roles are not played entirely by RIVET itself: it uses the HEPMC [48, 49] and YODA libraries for I/O and in-memory representation of events and histograms/analysis summary data. HEPMC events read into RIVET are wrapped into a more convenient `Rivet::Event` object, with a potential event-graph tidying step before storage as the event currently being analysed. The YODA library was developed primarily for use with RIVET, and has a similar layered structure with C++ and Python interfaces and user scripts, but is a general-purpose tool for statistics without particle-physics specialisations.

### Event loop

Internally, RIVET is primarily a framework for executing analysis routines on the incoming stream of events. The top-level structure of this framework in terms of code objects, user-facing scripts, and data flows is illustrated in Figure 1.

As with most such frameworks, this is broken into three phases: initialisation, execution, and finalisation. RIVET analysis objects, which inherit from the `Rivet::Analysis` base class, have three methods (functions), which are called at each stage: `init()` during initialisation, `analyze(const Rivet::Event&)` for each event during the execution loop, and an optional `finalize()` called once at the end of the run. Initialisation and finalisation are used for the set-up and pull-down phases of the analysis, most notably for creating histogram objects with appropriate types and binnings during initialisation using the `Analysis::book()` methods, and scaling or normalising them (or performing arbitrarily more complicated post-

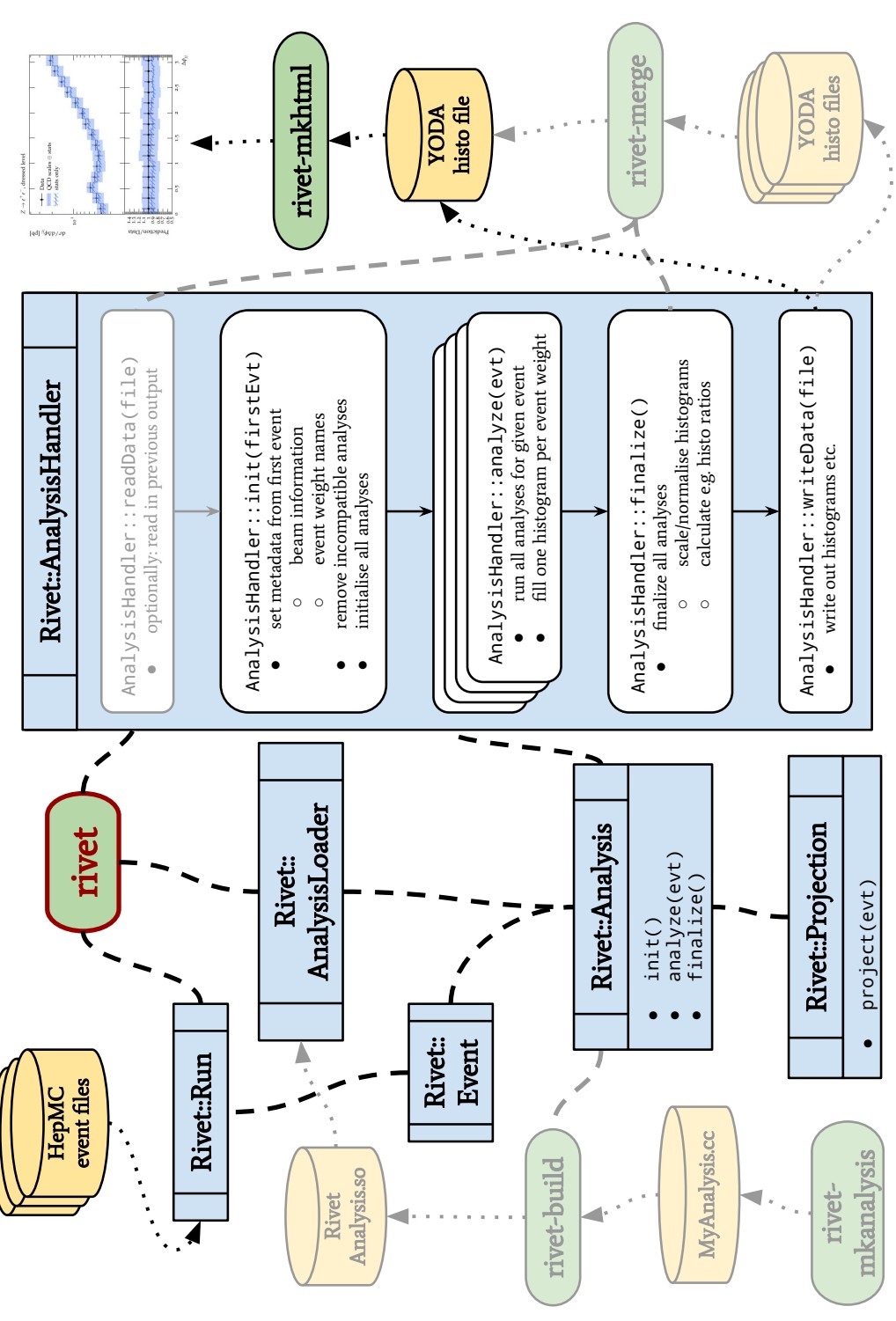

Figure 1: The RIVET system's main components and data flows. Library classes are shown in blue rectangles with functions in white, filesystem data in yellow cylinders, and user-interface scripts in green ovals. Dotted arrows indicate data flows, and dashed lines the main relationships between classes and scripts. Greyed components are optional paths for analysis preparation and post-processing.

processing manipulations and combinations) in the `finalize()` step. The execution step for each event involves computing physical quantities from the provided event, and using them to make control-flow decisions, and filling of histograms. However, experience shows that many calculations are common to the majority of analyses at a particular experiment or even whole collider: repeating such computations for each analysis does not scale well — as noted by the large experimental collaborations, which centralise the processing and calculation of common physics objects for both collider data and MC simulations. RIVET solves this problem by a semi-opaque method named "projections".

**Projections**

A projection is a stateful code object inheriting from the `Rivet::Projection` class, which provides a `project(const Rivet::Event&)` call signature. This operation computes a set of physical observables, e.g. the set of all final-state particles meeting certain kinematic or particle-ID requirements, an event-shape (e.g. eigenvalues of the sphericity tensor family [50]), or collections of collimated particle-jets. Each projection subclass adds custom functions by which these observables may be retrieved after computation. The strength of using projections as computational objects rather than basic functions, is that they can be stored: this permits caching, whereby a second (and third, and so-on) calculation of the same quantity from the same event automatically takes the short-cut of returning the previously calculated version.

In RIVET, this caching requires central storage of each projection, and a mechanism for determining whether two projections are equivalent. The latter is performed via a `SomeProj::compare(const Projection&)` method, specific to each projection type, and the former requires that projections are registered with the RIVET core in the initialisation phase of analysis execution. The `compare()` method typically compares numerical configuration parameters for the operations to be performed by the projection, e.g. the $p_T$ and $\eta$ cut values, or the clustering measure, radius, and grooming operations to be performed in jet reconstruction. It is guaranteed that projection comparisons are only ever performed between projection objects of the exact same type, via the C++ runtime type information (RTTI) system.

Registration is performed during initialisation by each analysis, using the `Analysis::declare()` method: this compares a configured projection object to all those currently known to the system, and either registers a *clone* of the new, distinct object in the global system with a character-string name specific to the registering analysis, or identifies an exactly equivalent projection and makes a new link to it via the same (name, analysis) pair. The use of cloning during projection declaration means that memory and pointer management problems are avoided, and the analysis-authoring user can focus on the physics-logic of their code, rather than C++ technicalities.

The power of projections is greatly enhanced by the ability to "chain" them arbitrarily deep: any projection can itself register a set of its own projections, like standard building blocks to construct a well-behaved calculation. The same declaration and cloning abilities are available to projections, with the only additional requirement on a projection-authoring user being that they include calls to their contained projections' `compare()` methods via the convenience `pcmp()` method.

By the end of initialisation, all projections should be declared and uniquely registered to the RIVET core, and are known to each analysis (or chaining projection) by a string, e.g. `"Jets"`, or `"CentralMuons"`, or `"Thrust"`. They may then be called by that string name using the projection/analysis' `apply<T>()` method. This is unfortunately complicated by C++'s unawareness of the exact type, so the user must specify the projection type `T` that they want the returned object to be. The `auto` keyword in C++11 makes this operation slightly simpler, and overall the configuration and calling of projections is a simple operation that automatically

provides result-caching for common quantities, as well as encapsulating a lot of standard detail and workarounds for misconceptions and problems in computation of physical quantities from HEPMC/RIVET events.

**Event analysis**

During the execution phase, both analyses and projections can also use more direct physics calculation tools. The most important of these are the `Particle`, `Jet` and `FourMomentum` classes, and an array of functions for computing kinematics or particle ID properties, e.g. a set of `deltaR()`, `mT()`, `isCharged()`, `hasCharm` etc. functions which are also largely replicated on the class interfaces of `Particle`, `Jet` and `FourMomentum` themselves. From RIVET 2.6 onwards, the `Particle` class acquired the ability to recursively contain more `Particles`, useful for definition of reconstructed "pseudoparticle" objects like $Z$ bosons, reconstructed top-quarks, and photon-dressed charged leptons. This development also brings particles and jets conceptually closer together and indeed many useful functions (e.g. ancestor and descendent checks) are provided for general application to their common base class, `ParticleBase`. The two concepts are not yet fully unified, however, with `Particle` but not `Jet` providing particle-ID information, and `Jet` but not `Particle` supporting truth-tagging using ghost-associated $c$ and $b$ hadrons, and $\tau$ leptons.

Particles and jets are returned by their respective "finder" projections as lists, known as `Particles` and `Jets` containers, which are often sorted in decreasing $p_T$. Picking the leading objects from such vectors is hence easy using standard vector indexing, but sometimes more complex operations are needed, such as filtering out objects that do or don't meet certain criteria, or summing the $p_T$s of all the objects in a list to compute an $H_T$ or $m_{\text{eff}}$ measure. RIVET makes use of modern "functional" programming capabilities in C++ for this, in the form of `select()` and `discard()` functions, which take a list and a "functor" (function object) as arguments: the function then returns a new list containing the physics objects from the first list which caused the functor to return true or false, depending on whether the logic is to have selected or discarded those which returned true. This system is immensely flexible and far more convenient than standard-library tools, and will be described later in more detail, with examples. Particle and jet lists may also be concatenated using the `+` operator, unlike STL vectors.

**Analysis-routine compilation & loading**

Analyses are compiled separately from the core RIVET library, as "plugin" shared libraries which are loaded explicitly by searching the internal analysis path (set either programmatically or via the `RIVET_ANALYSIS_PATH` environment variable) for libraries matching the pattern `Rivet*.so`. A `rivet-build` script is supplied as a plugin-building frontend for the C++ compiler, both for simplicity and to ensure exact compatibility of compiler options with the core library. This script can build multiple analyses into a single plugin, and as well as for user convenience is used to build the more than 700 included analysis codes into plugins grouped by experiment or collider.

## 3 New features

### 3.1 Combineable kinematic cuts and filtering functors

Clarity and expressiveness of analysis logic are key to the RIVET design philosophy, and continuous development of the RIVET API has refined how this goal has been technically achieved.

Ideal demonstrations of this have been the addition of major code systems for more configurable expression of kinematic cuts and other physics-object filtering, as well as cosmetic reduction of clutter such as

- replacement of the rather cumbersome `addProjection()` and `applyProjection()` functions with neater `declare()` and `apply()` names;

- direct provision of momentum-like properties on `Particle` and `Jet` types without needing to retrieve the contained `FourMomentum`;

- automatic implicit casting of `Jet` and `Particle` to `FourMomentum` to functions expecting the latter as an argument, and similar implicit casting between FASTJET and RIVET jet types; and

- provision of `abseta()`, `absrap()` and `abspid()` methods on physics objects to reduce the parenthetic noise introduced by user calls to `abs()` or `fabs()`.

Here we summarise first the `Cuts` system, and then its extension to filtering functors.

In the original API, many functions used lists of several, perhaps optional, floating-point valued arguments, e.g. `FinalState(double etamin, double etamax, double ptmin)` or `fs.particles(double ptmin)`. Not only were these inflexible, not allowing "inline" cuts other than the hard-coded ones, and sometimes annoyingly verbose — as with nearly always symmetric rapidity cuts, or often having to supply `DBL_MAX` values to apply a $p_T$ cut only — but at the compiled code level they were ambiguous. It was easy to forget if the $\eta$ cuts or the $p_T$ cut came first, and accidentally require something like $\eta > 10$, with clear consequences: the compiler has no way of telling one `double` from another and warning the analysis author of their error.

To address this without adding surprising behaviours, myriad specially named functions, or other undue complexity to analysis code, we developed a `Cut` object based on C++11 smart pointers, with function overloads allowing `Cut`s to be combined using any normal boolean operators, e.g. `operator(const Cut&, const Cut&)` → `Cut`. Several specialisations of `Cut` are provided in the `Rivet/Tools/Cuts.hh` header, providing `enum`s which map to cuts on $p_T$, rapidity, pseudorapidity, energy, $E_T$, charge, etc. These can be applied on `FourMomentum`, `Particle` and `Jet` objects, and for `Particle` a `Cuts::pid` enum is available in addition, supporting equality as well as inequality comparisons and for use with particle-ID `enum`s like `PID::ELECTRON`. This PID cut, as well as the rapidities and charges, is also available in "abs" form i.e. `Cuts::abspid`, `Cuts::abseta`, etc., to allow clearer and more compact analysis cut expressions. The use of explicit cut expressions like `fj.jets(Cuts::pT > 20*GeV && Cuts::absrap < 2.5)` has made RIVET analysis code both easier to write and to read.

However, these cuts are still static, predefined entities: useful for the majority of object selection requirements, but not all. In particular, the cuts are defined for one object at a time, while an important class of analysis cuts select or reject objects depending on their relations to other objects in the event, e.g. lepton–jet isolation or overlap-removal. The logic of such isolations is not particularly complex, but typically involves several nested `for`-loops, which require reverse-engineering by a reader wishing to understand the analysis logic. In addition, the C++ Standard Template Library interface for deleting objects from containers — the so-called "erase–remove idiom" — is verbose and non-obvious, distracting from the physics goal of object filtering. For this reason, the `(i)filter` functions already described were added, as well as event higher-level functions like the `(i)select/discardIfAny` set, which accept *two* containers and a comparison functor, for e.g. implicitly looping over all leptons and in-place discarding any jet that overlaps with any of them.

These have revolutionised the writing of isolation-type code, reducing complex multi-loop code to one-line calls to a filtering function, leveraging the substantial library of function objects. Use of the C++11 `std::function` interface means that these functors can be normal functions, e.g. the `isCharged(const Particle&)` utility function, but powerfully they may also be stateful objects to implement configurable cuts such as `pTGtr(const FourMomentum &, double)`, or `hasBTag(const Cut&)`. This latter class allows the function to be defined with respect to another object, selected "live" in the body of a loop, and even *defined* inline, using the C++11 anonymous "lambda-function" system.

Similar uses of the functional coding paradigm in C++11 have led to other classes of functor for sorting (which may be passed to functions like `ParticleFinder::particles()` or `JetFinder::jets()`), and for computation of variables (for use with functions such as `sum( const Particles&)`). Again, these can accept functions or function objects, including C++ inline lambda functions. With care, such interface evolutions have provided a step-change in the power and expressiveness (and compactness) of RIVET analysis logic, not just without sacrificing readability and self-documentation, but improving it.

## 3.2 Systematic event weights

One of the main structural changes in RIVET 3 is the handling of event weights. In the quest of understanding better the uncertainties in the event generator modelling, it is important to vary scales, parameters, parton density functions, etc. Previously, this was handled by generating many event samples corresponding to different settings. Today most event generators instead accomplish the same thing by assigning a set of different weights for each generated event corresponding e.g. to different scale choices.

Another aspect of weights is that in some next-to-leading order (NLO) QCD generators there is a system of combining events into event groups. In e.g. a dipole subtraction scheme [51], real emission events would be accompanied by a set of counter events with Born level kinematics corresponding to the possible dipole mappings of the real emission phase space. The real emission event would be weighted according to the tree-level cross section, while the counter events would be negatively weighted according to the corresponding Born cross section times the releavnt dipole splitting. This means that great care must be taken when analysing these events and filling a histogram with the weights. For well behaved (soft and collinearly safe) observables the real and counter events will normally end up in the same bin. But due to the different underlying kinematics it may happen that they don't. Also, to obtain the correct statistical uncertainty, a histogram bin cannot be filled once with each (sub-)event. Instead, the bin should be filled once for each such *event group*, with a weight given by the sum of the weights of the real and counter-events.

Finally, matching the ever increasing luminosity of today's colliders often requires the generation of extremely large event samples. To make this efficient, the generation is often divided in to many parallel runs that need to be combined afterwards, treating the weights in a statistically correct way. To make sure that the weights are treated correctly in all such cases, the weight handling is no longer directly exposed to the implementer of an analysis, but is handled behind the scenes, as described in the following.

### 3.2.1 Handling of multiple event weights

To avoid having to run each analysis many times for the same event, i.e. once for each supplied event weight, the weight handling is no longer directly exposed to the `Analysis` class. This means that the histograms and other analysis objects to be filled in an analysis are encapsulated in a wrapper class actually containing several copies of the same histogram, one for each event weight. For the analysis implementer things look more or less as in RIVET 2. The histograms

are still handled with pointers, `Histo1DPtr`, but while before these were standard shared pointers, they are now much more sophisticated.

The main visible changes are in the booking and the filling of histograms. While before a histogram would be booked as `hist = bookHisto1D(...)`, the syntax has changed to `book(hist, ...)` (and similarly for the other analysis object types). In addition, rather than always having to explicitly fill a histogram with a weight in RIVET 2 as `hist->fill(x,weight)`, the new scheme will handle the weights separately changing this syntax to[1] `hist->fill(x)`.

What happens behind the scenes is that `book(...)` will actually create several YODA histograms, one for each weight. This means that it is not possible at this point in the `Analysis::init()` function to actually access a particular instance of the histogram through the `Histo1DPtr` pointer. In the same way in the `Analysis::analyze(...)` function it is only possible to fill the histograms using the `fill(...)` function, while any other attempt to manipulate a particular histogram will fail. Calling the `fill(...)` function will actually not directly fill each histogram with the corresponding event weight, rather the fills will be recorded in a list, and the actual histogram filling will only take place after all analysis of an event (group) has been completed. The reason for this will become clear in Section 3.2.2 below.

At the end of a run, `Analysis::finalize()` is called for all analyses, once for each event weight. The syntax in this function is completely unchanged from RIVET 2, and in each call the `Histo1DPtr` will work as a normal pointer to a `YODA::Hist1D` object, that can be manipulated in the normal way.

It is worth noting that the implementer typically does not need to worry about the event weight when writing code. This of course assumes that the user is not expected to fill a histogram with a combination of different event weights. Such an event-weight manipulation is better handled within the actual generators, and hence the generators are expected to produce self-contained event weights ready for histogram filling, with the exception of counter-events discussed in the next section.

The weights are taken from the input HEPMC file, where they must be given a name. There is so far no standard format for the weight names, and basically any character string can be handled by RIVET. It should be noted, however, that names of analysis objects in the output YODA file will have the corresponding weight names appended, enclosed in square brackets, and in general it is not advisable to use special characters in the weight names. In addition RIVET will treat one of the weight supplied in the HEPMC as *nominal*, and the corresponding analysis objects will be stored in the output YODA file without the weight name appended. Also for the nominal weight, there is no fixed convention for the name, and RIVET will assume that a weight named `""` (empty string), `"0"`, `"Default"` or `"Weight"` is the nominal one. If there is no weight with such a name, the first weight found will be treated as nominal.

Handlers are provided in the Python interface to extract or strip the weight names from histogram paths.

### 3.2.2 Handling of NLO events and counter-events

When handling an event group of NLO-generated real emission events with corresponding counter-events, the events are fully correlated and it is important that each event group is treated as one, so that each histogram fill is given by the sum of the event weights in the group. In addition these fills should have the correct error propagation encoded in the sum of weights (SoW) and the sum of the squared weights (SoSW).

---

[1] A second argument can still be supplied here, but this is then interpreted as a number that should be multiplied by the event weight

The idea is that in a histogram of a soft- and collinear-safe observable, the real emission event and the corresponding counter-events will always end up in the same bin in the limit when the real emission is soft and/or collinear. In this limit, the weight for the real event approaches positive infinity and the weight of one or more of the counter-events approaches negative infinity. However, there is always a possibility that a fill of the real event ends up very close to a bin edge while a counter-event ends up on the other side of the edge, ruining the NLO cancellation. In RIVET the trick to solve this problem is to not just fill at the given value of the observable, but to spread it out in window, possibly filling also adjacent bins.

The full procedure to handle this with correct error propagation is fairly complicated, and is completely hidden from the user and analysis implementer. For reference the full description if the procedure is given in Appendix A.

### 3.2.3 Event-weight analysis

In recent years it has become increasingly important to study the event-weight distribution directly. Producing very large event samples can quickly become an expensive endeavour when the overall sample size is determined by the delicate trade-off between the desired statistical precision and how much of the CPU budget is being spent simulating events with negative event weights, which would ultimately reduce the statistical power of the sample. For unweighted events, both the spread of the event weights as well as the fraction of negative weights in the sample need to be understood in order to be able to project the CPU cost of the sample correctly. Although RIVET 3 has put a lot of effort into hiding the handling of the event weights from the user, it is still possible to retrieve them and treat them as an observable when filling a histogram. The corresponding syntax is illustrated in the routine `MC_XS`. Note that this will not work if the sample makes use of counter-events.

### 3.2.4 Re-entrant finalize and run merging

It is quite common to divide up RIVET analyses of very large event samples in smaller runs, and merge the produced YODA files. Previously this has been done with the `yodamerge` Python script distributed with YODA. However, this script was completely ignorant of the way the different analysis objects were produced. In order to better handle this RIVET has since version 2.7 introduced the Python script `rivet-merge` which does the same thing but using the complete knowledge of how the analysis objects were filled in `Analysis::analyze(...)` and manipulated in `Analysis::finalize()`.

The way this is implemented means that all analysis objects come in two different instances. One is called *raw* and is used only for filling in the `Analysis::analyze(...)` functions. Before a call to `Analysis::finalize()` the raw analysis objects are copied to the *final* instances which are then used for manipulating the objects into their final form to be plotted. In this way the `Analysis::finalize()` function can be run several times and is then called *re-entrant*. The user will notice that the output YODA file contains both sets of analysis objects, one with the standard naming, and one with the same name prefixed by `/RAW/`. [2]

In this way the `rivet-merge` script can read in YODA files, create and run the `Analysis::init()` for the corresponding analyses, merge all raw analysis objects together and run `Analysis::finalize()`. When doing this it is important to note that there are two different kinds of merging possible. In one situation we have run RIVET on several completely equivalent event samples, and in the other the runs have been on different kinds of event samples which should be combined in another way. An example of the latter is when the same analysis is

---

[2] Handlers are provided in the Python interface to identify these histograms when reading YODA files for post-processing.

run on event samples with different $\sqrt{s}$, and in the end will produce e.g. ratios of distributions between different energies.

To understand the differences between these two merging schemes we assume we have a number of runs, each with the cross-section $\sigma_i$ as reported by the generator, the total sum of weights $S_{w,i}$, and a number of "raw" histogram bins each with a sum of weight $S_{w,i}^{[b]}$ and the sum of squared weights $S_{w^2,i}^{[b]}$. When finalized the histogram bin will typically have a cross-section $\sigma_i^{[b]} = \sigma_i S_{w,i}^{[b]}/S_{w,i}$ with an estimated error of $\delta\sigma_i^{[b]} = \sigma_i \sqrt{S_{w^2,i}^{[b]}}/S_{w,i}$. But we can also imagine other situations where the final plot point is a fraction of the total $r_i^{[b]} = \sigma_i^{[b]}/\sigma_i$, or if we want to be a bit more general any ratio of cross-sections $r_i^{[b/a]} = \sigma_i^{[b]}/\sigma_i^{[a]}$. We note that for uniformly weighted events an individual generator run corresponds to an integrated luminosity $\mathcal{L}_i = S_{w,i}/\sigma_i$.

Now if the runs to be merged have exactly the same process, the weights in the combined histogram will simply be the added weights,

$$S_w = \sum_i S_{w,i}, \quad S_w^{[b]} = \sum_i S_{w,i}^{[b]}, \quad \text{and} \quad S_{w^2}^{[b]} = \sum_i S_{w^2,i}^{[b]} \tag{1}$$

and the cross-section for the combined files will be a weighted average

$$\sigma \;=\; \frac{1}{S_w} \sum_i S_{w,i} \sigma_i \,.$$

Alternatively, to be on the safe side, if the files have identical processes, but different weight variations, we might want to use the effective number of entries, $\mathcal{N} = S_w^2/S_{w^2}$, as weights,

$$\sigma = \frac{1}{\mathcal{N}} \sum_i \mathcal{N}_i \sigma_i \quad \text{and} \quad \delta\sigma^2 = \frac{1}{\mathcal{N}^2} \sum_i \mathcal{N}_i^2 \delta\sigma_i^2 \,. \tag{2}$$

For each bin we will e.g. have the plot value $\sigma^{[b]} = \sigma S_w^{[b]}/S_w$ with an estimated error of $\delta\sigma^{[b]} = \sigma \sqrt{S_{w^2}^{[b]}}/S_w$ as for the individual histograms.

Turning now to the case of adding histograms with different processes, the case where the histograms are already normalised to cross-section is the easiest, since we can then simply add

$$\sigma = \sum_i \sigma_i, \quad \sigma^{[b]} = \sum_i \sigma_i^{[b]}, \quad \text{and} \quad \delta\sigma^{[b]} = \sqrt{\sum_i \left(\delta\sigma_i^{[b]}\right)^2} \,. \tag{3}$$

For adding the raw histograms we need to expand out the cross-sections in terms of weights,

$$\sigma \frac{S_w^{[b]}}{S_w} = \sum_i \sigma_i \frac{S_{w,i}^{[b]}}{S_{w,i}}, \quad \text{and} \quad \delta\sigma^2 \frac{S_{w^2}^{[b]}}{S_w^2} = \sum_i \frac{\delta\sigma_i^2 S_{w^2,i}^{[b]}}{S_{w,i}^2} \,.$$

In other words, the ratio of the weights to the total is a cross-section-weighted average, and we can write

$$S_w^{[b]} = \frac{S_w}{\sigma} \sum_i \sigma_i \frac{S_{w,i}^{[b]}}{S_{w,i}} \quad \text{and} \quad S_{w^2}^{[b]} = \frac{S_w^2}{\sigma^2} \sum_i \frac{\sigma_i^2 S_{w^2,i}^{[b]}}{S_{w,i}^2} \,. \tag{4}$$

However, the $S_w$ is arbitrary (two equations and three unknowns above), and this is related to the fact that the combined histograms no longer necessarily corresponds to a particular integrated luminosity. This, in turn, means that it is not possible to first combine histograms of different processes and then combine these with others of identical combinations.

If the different runs do correspond to the same integrated luminosity, of course the combined run should correspond to the same. One reasonable way of obtaining this could be to let the integrated luminosity for the merged sample be the cross-section-weighted average of the individual samples,

$$\frac{S_w}{\sigma} = \mathcal{L} = \frac{1}{\sigma}\sum_i \sigma_i \mathcal{L}_i = \frac{\sum_i S_{w,i}}{\sigma}. \tag{5}$$

In conclusion, the way `rivet-merge` combines raw histograms for different runs is for different processes (the default)

$$S_w^{[b]} = \frac{S_w}{\sigma}\sum_i \sigma_i \frac{S_{w,i}^{[b]}}{S_{w,i}} \quad \text{and} \quad S_{w^2}^{[b]} = \frac{S_w^2}{\sigma^2}\sum_i \frac{\sigma_i^2 S_{w^2,i}^{[b]}}{S_{w,i}^2}, \tag{6}$$

while for combining histograms with identical processes (using the command argument `-e` or `--equiv`)

$$S_w^{[b]} = \sum_i S_{w,i}^{[b]} \quad \text{and} \quad S_{w^2}^{[b]} = \sum_i S_{w^2,i}^{[b]}. \tag{7}$$

Similarly the resulting YODA file will have

$$S_w = \sum_i S_{w,i} \quad \text{and} \quad S_w^2 = \sum_i S_{w,i}^2 \tag{8}$$

in all cases, while for identical processes

$$\sigma = \frac{1}{\mathcal{N}}\sum_i \mathcal{N}_i \sigma_i \quad \text{and} \quad \delta\sigma^2 = \frac{1}{\mathcal{N}^2}\sum_i \mathcal{N}_i^2 \delta\sigma_i^2, \tag{9}$$

and for different processes

$$\sigma = \sum_i \sigma_i \quad \text{and} \quad \delta\sigma^2 = \sum_i \delta\sigma_i^2. \tag{10}$$

It is important to note that not all analyses in RIVET necessarily have re-entrant `Analysis::finalize` methods. Work is in progress to convert them all, but it is not yet finished. The ones that have been done are given the status `REENTRANT` in the `.info` file. The requirements that need to be met for this status are as follows:

- All information that is needed to `finalize` an analysis *must* be encoded in properly booked analysis objects.

- *All* analysis objects that can be used in an analysis must be properly booked in the `Analysis::init()`, also those that are not needed in the current run (e.g. if the analysis can be run at different $\sqrt{s}$).

- Non-fillable analysis objects (such as `Scatter2D`) cannot be merged automatically. These are normally only constructed in `Analysis::finalize()` as e.g the result of dividing two histograms, and it is recommended to book these analysis objects in the `Analysis::finalize()` method rather than in `Analysis::init()`.

An example analysis with the `REENTRANT` status is `ALICE_2012_I930312`.

### 3.3 Heavy ion physics

The RIVET framework as such, has no preference for any type of collision system, as long as simulated collider events by a Monte Carlo event generator can be represented in the HEPMC format. The possibility to add experimental analysis plugins based on data from heavy ion experiments is as such not new, and was for instance used in the implementation of the analysis `LHCF_2016_I1385877` [52].

The possibility of having a heavy ion beam is, however, not sufficient to implement the bulk of existing experimental analyses, as the employed techniques differ from standard techniques in proton-proton and electron-positron collisions. The threshold for implementing even simple analyses has thus previously been too high for any real progress to be made into this area. From RIVET 2.7.1 onward, new `Projection`s and other tools to facilitate the implementation of heavy ion analyses have been added [53], and carried through to version 3. New features include:

- A designated centrality framework, outlined in Section 3.3.1.

- A framework for calculating flow observables, based on the Generic Framework [54,55], outlined in Section 3.3.2.

- Designated `PrimaryParticle` projections for implementing experimental definitions of primary and secondary particles.

- Re-entrant finalization (see Section 3.2.4 for a separate introduction) to allow for heavy ion to $pp$ ratio figures, such as nuclear modification factors $R_{AA}$, but also useful for statistically correct merging in general.

- Pre-loading of calibration data and analysis options (see Section 3.7 for a separate introduction) to allow for centrality selection, but also useful in several other cases.

- An `EventMixingFinalState` projection to allow Monte Carlo generated events to be mixed, to allow access to particles from distinct events in order to correct correlation functions for the effects from limited acceptance and single particle distributions.

The technical use of the centrality framework and the framework for flow observables, is outlined in Sections 3.3.1 and 3.3.2 respectively. For a more complete overview of all new methods, as well as a physics introduction, the reader is referred to Ref. [53]. The tools introduced for heavy-ion physics, are not limited in use to analyses of heavy-ion beams. Already a number of analyses of $pp$ collisions implementing such techniques are made available. A full list of all currently available analyses either implementing heavy ion functionality, or containing heavy ion beams, is given in Table 1.

#### 3.3.1 Centrality estimation

The size and transverse shape of the interaction region is of particular interest in the analyses of colliding nuclei, but cannot be measured directly in experiments. Experiments instead classify collisions according to a single event observable $N$, defining the centrality in percentiles of the distribution $d\sigma_{\mathrm{inel}}/dN$, such that the centrality of a collision is:

$$c = \frac{1}{\sigma_{\mathrm{inel}}} \int_N^\infty \frac{d\sigma_{\mathrm{inel}}}{dN'} dN'. \tag{11}$$

The single event observable $N$ can then be defined in one of three ways:

Table 1: All RIVET analyses implementing one or more heavy ion features, or heavy ion beams. An up-to-date list can be found at `https://rivet.hepforge.org`.

| Analysis name | System | Validated | Heavy ion features | Reference |
|---|---|---|---|---|
| ALICE_2010_I880049 | PbPb | Yes | centrality, primary particles, | [56] |
| ALICE_2012_I930312 | PbPb | Yes | centrality, heavy ion container, re-entrant finalize | [57] |
| ALICE_2012_I1127497 | PbPb | Yes | centrality, heavy ion container, re-entrant finalize | [58] |
| ALICE_2012_I1126966 | PbPb | No | centrality, primary particles | [59] |
| ALICE_2013_I1225979 | PbPb | No | centrality, primary particles | [60] |
| ALICE_2014_I1243865 | PbPb | No | centrality, primary particles | [61] |
| ALICE_2014_I1244523 | $p$Pb | No | centrality, primary particles | [62] |
| ALICE_2016_I1394676 | PbPb | No | centrality, primary particles | [63] |
| ALICE_2016_I1419244 | PbPb | No | centrality, generic framework | [64] |
| ALICE_2016_I1507090 | PbPb | No | centrality, primary particles | [65] |
| ALICE_2016_I1507157 | $pp$ | No | event mixing | [66] |
| ATLAS_2015_I1360290 | PbPb | No | centrality | [67] |
| ATLAS_2015_I1386475 | $p$Pb | No | centrality | [68] |
| ATLAS_PBPB_CENTRALITY | PbPb | No | centrality | [67] |
| ATLAS_pPb_Calib | $p$Pb | No | centrality | [68] |
| BRAHMS_2004_I647076 | AuAu | No | centrality, primary particles | [69] |
| CMS_2017_I1471287 | $pp$ | No | generic framework | [70] |
| LHCF_2016_I138587 | $p$Pb | No | — | [52] |
| STAR_2016_I1414638 | AuAu | No | centrality | [71] |

- As the measured distribution by the experiment, translating the percentile cuts directly to cuts in a measured observable. This is of course the definition most in line with the RIVET philosophy, but is not always feasible.

- In a similar way as the experiment, but using the Monte Carlo generator to generate $\mathrm{d}\sigma_{\mathrm{inel}}/\mathrm{d}N$, defining the percentile cuts.

- Using a model's impact parameter ($b$) in place of $N$, thus comparing a theoretical centrality to the measured one.

In experiments, $N$ is often chosen to be an observable proportional to particle production in the forward region. Since the ability of a given Monte Carlo generator to reproduce this specific observable should not be a limiting factor, the two latter options have been added. In such cases, the distribution $\mathrm{d}\sigma_{\mathrm{inel}}/\mathrm{d}N$ must be known before the execution loop is initiated, i.e. when the method `analyze(const Rivet::Event&)` is called for the first time. To that end, a calibration run using a special calibration analysis must be performed.

The calibration analysis is a simple analysis with the sole purpose of filling histograms containing the distributions $1/\sigma_{\mathrm{inel}}\,\mathrm{d}\sigma_{\mathrm{inel}}/\mathrm{d}N$ and $1/\sigma_{\mathrm{inel}}\,\mathrm{d}\sigma_{\mathrm{inel}}/\mathrm{d}b$. The output from running this analysis is read in using the `--preload` option. This option reads the filled histogram objects into RIVET, and makes them accessible for the duration of the (second) run. A `CentralityProjection` can then be booked by calling `declareCentrality(const SingleValueProjection &proj, string calAnaName, string calHistName, const string projName)`. Here `proj` is a projection returning the current value of $N$, `calAnaName` is the name of the calibration analysis, `calHistName` is the name of the calibration histogram and `projName` is the given name of the centrality projection. In the execution loop, the projection can be applied to the current event, and the method `cent()` will return $c$ for the current event.

The user can select between the above mentioned centrality definitions at runtime, using an analysis option (see Section 3.7). The analysis option `cent=REF` (default) selects the measured distribution, `cent=GEN` selects the generated version of the measured distribution,

`cent=IMP` the impact parameter distribution and finally `cent=USR` allows the user to use a hard coded centrality value from the HEPMC input file (from HEPMC 3.0).

### 3.3.2  Flow measurements

A large subset of analyses of high energy heavy ion collisions, are concerned with studies of the azimuthal anisotropy of particle production. This is quantified in flow coefficients $v_n$'s, defined as the Fourier expansion of the particle yield with respect to the event reaction plane $\Psi_n$:

$$E\frac{\mathrm{d}^3 N}{\mathrm{d}^3 p} = \frac{1}{2\pi}\frac{\mathrm{d}^2 N}{p_\perp \mathrm{d}p_\perp \mathrm{d}y}\left(1 + 2\sum_{n=1}^{\infty} v_n \cos([n(\phi - \Psi_n)])\right). \tag{12}$$

Here $E$, $p_\perp$, $\phi$ and $y$ denote the particle energy, transverse momentum, azimuthal angle and rapidity, respectively. Since the reaction plane is not accessible experimentally, flow coefficients are often estimated from two- or multi-particle correlations. In RIVET we have implemented the Generic Framework formalism [54, 55], plus a number of convenient shorthands. The framework allows for quick evaluation of multi-particle correlations in terms of $Q$-vectors:

$$Q_n = \sum_{k=1}^{M} \exp(in\phi_k), \tag{13}$$

for an event of $M$ particles. Since a $Q$-vector requires just a single loop over data, as opposed to $m$ loops for an $m$-particle correlation, the Generic Framework reduces the computational complexity of multi-particle correlation analyses from $\mathcal{O}(M^m)$ to at most $\mathcal{O}(M\log(M))$. For a more thorough introduction to the Generic Framework, the reader is referred to the dedicated paper on heavy ion functionalities in RIVET [53], as well as Refs. [54, 55]. The following text will be mainly concerned with the technical usage in analysis code. In general, the Generic Framework expresses flow coefficients of $n$'th order in terms of $m$-particle cumulants of $n$'th order, denoted $c_n\{m\}$. Cumulants are again expressed as correlators of even order $\langle\langle m\rangle\rangle_{n_1,n_2,\ldots,-n_{m/2},\ldots,-n_m}$, which can finally be expressed algorithmically in terms of $Q$-vectors.

In order to access the Generic Framework functionality in RIVET for calculation of cumulants, the analysis must inherit from the `CumulantAnalysis` class, which itself inherits from the `Analysis` base class. This allows for $m,n$-correlators to be booked with a call to the templated method `template<unsigned int N, unsigned int M> bookECorrelator (const string name, const Scatter2DPtr hIn)`. Here the template arguments corresponds to $m$ and $n$, `name` is the given name of the correlator and `hIn` should contain the binning of the correlator (usually imported from the analysis `.yoda` file). Also available, is a `Correlators` projection, which is declared using the constructor `Correlators(const ParticleFinder& fsp, int nMaxIn, int pMaxIn)` (also available in a $p_\perp$ binned version). Here `fsp` is an already declared `ParticleFinder` derived projection from which particles should be drawn, `nMaxIn` is the maximal sum of harmonics to be generated (e.g. 4 for $c_2\{2\}$) and `pMaxIn` the maximal number of particles to be correlated. If all desired correlators for a given analysis are already booked, the two maximal values can be extracted automatically from booked correlators by calling `getMaxValues()`, which returns a pair of `int`s, where the first is `nMaxIn` and the second is `pMaxIn`.

In the `analyze` step of the analysis, correlators can be filled with an applied `Correlators` projection. The projection is applied as usual e.g. by `const Correlators& c = apply< Correlators>(event, "ProjectionName");`, and a booked $m,n$-correlator is filled as `corrPtr->fill(c);`.

In the `finalize` step of the analysis, correlators can be cast into cumulants or flow coefficients. If an analysis implements e.g. experimental data on integrated $c_2\{2\}$ and $v_2\{2\}$, the

methods `cnTwoInt(Scatter2DPtr h, ECorrPtr e2)` and `vnTwoInt(Scatter2DPtr h, e2)` maps the correlator `e2` to scatters pointed to by `h`.

## 3.4 Deep-inelastic *ep* scattering and photoproduction

Although RIVET traces its conceptual origins directly back to HZTool [72,73], a Fortran package developed by the H1 and ZEUS collaborations at the HERA *ep* collider to facilitate the comparison of their measurements to MC predictions and each other, rather few deep inelastic scattering (DIS) or photoproduction measurements have been implemented in RIVET to date. This is partly because of the existing and extensive library of such analyses in HZTool. Such measurements contain important and unique information, principally on high-energy QCD dynamics and hadronic structure, which remains relevant to current and future analyses. The need to preserve access for use with modern event generators, and to exploit the ongoing benefits of new development in RIVET— several of which are informed by lessons learned from HZTool— has grown over the years, and has been further stimulated by the active community work toward a future electron–ion collider. As a consequence efforts have been made to interface the old HZTool Fortran routines to RIVET, and a proper plug-in library is all but released [74]. In parallel to this the latest version of RIVET contains a few more HERA analyses, but more importantly it provides a series of new projections to extract common DIS and photoproduction event properties, greatly facilitating the addition of new (or indeed old) DIS and photoproduction analyses in future. The currently available HERA routines are given in Table 2.

### 3.4.1 Kinematic definitions

In following the RIVET philosophy of defining observable in terms of final state particles, to avoid model- and generator-dependence, a number of physics issues arise which were in general noted, but not solved, in HZTool. In particular, it is not always clear from the H1 and ZEUS analyses how to do the following, without examining the (unphysical and generator-dependent) parton-level event history.

**Identifying the scattered lepton**     In most DIS events, there is a single obvious scattered lepton candidate. However, other leptons may be present in the event and RIVET (and in principle physics!) require a way resolving any ambiguity based upon observable information — essentially the kinematics. Unfortunately most HERA publications do not provide this information (and indeed were often corrected to the MC-dependent electron vertex). The `DISLepton` projection therefore provides a few of pragmatic recipes to identify the scattered lepton, with options to select the highest energy lepton (default) or by rapidity or transverse energy. The electron may also be required to be prompt (default).

The scattered neutrino kinematics in charged current events should be determined from the missing transverse energy. However, no such analyses are currently implemented.

**Treating electroweak corrections to the Born-level process**     Many HERA analyses were corrected to the "Born" level, again leaving some ambiguity about how radiated photons should be treated, when they are present. Of course, events may be run at fixed order with QED radiation turned off, and, while model-dependent in principle, this is most likely the closest approximation to what was done in the original measurement. To allow the study of such effects, the `DISLepton` projection will, if requested, return the kinematics of the electron including in the lepton four-momentum all photons within some cone (thus recovering final-state QED radiation to some approximation), and excluding from the energy of the beam

Table 2: All RIVET analyses routines available in a reasonable state of usability (though in general not formally validated). An up-to-date list can be found at https://rivet.hepforge.org.

| Analysis name | System | Experiment | Measurement features | Reference |
|---|---|---|---|---|
| H1_1994_S2919893 | $e^{\pm}p$ DIS | H1 | Energy flow and charged particle spectra | [75] |
| H1_1995_S3167097 | $e^{\pm}p$ DIS | H1 | Transverse energy and forward jet, low-$x$ | [76] |
| H1_2000_S4129130 | $e^{\pm}p$ DIS | H1 | Energy flow | [77] |
| H1_2007_I746380 | $e^{\pm}p$ DIS & | | | |
| | $ep \to \gamma p$ | H1 | Diffractive production of dijets | [78] |
| H1_2015_I1343110 | $e^{\pm}p$ DIS | H1 | Diffractive dijets | [79] |
| HERA_2015_I1353667 | $e^{\pm}p$ DIS | H1& ZEUS | Combined H1/ZEUS $D^*$ production | [80] |
| ZEUS_2001_S4815815 | $ep \to \gamma p$ | ZEUS | Dijets | [81] |
| ZEUS_2008_I763404 | $ep \to \gamma p$ | ZEUS | Diffractive photoproduction of dijets | [82] |
| ZEUS_2012_I1116258 | $ep \to \gamma p$ | ZEUS | Inclusive jet cross sections | [83] |

electron all photons in some cone (this accounting at some level initial state radiation). A hadronic isolation criterion may also be applied.

The `DISKinematics` projection then calculates the usual DIS variables, such as Björken $x$, $y$ and $Q^2$, from the identified scattered electron and beam energy. The `DISFinalState` returns the final state particles excluding the scattered lepton, optionally boosted into the hadronic centre-of-mass frame, the Breit frame, or left in the laboratory frame.

**Identifying the photon kinematics in photoproduction**  Photoproduction is viewed as a special, low-$Q^2$ case of DIS. In most analyses, a veto is applied on the scattered electron entering the detector acceptance, typically corresponding to an upper cut on $Q^2$ of 1–4 GeV$^2$. The `DISKinematics` projection may thus be used to obtain the energy and virtuality of the interacting photon.

**Defining diffraction and its kinematics**  For diffractive analyses with a tagged forward proton, the issues are similar in principle to those associated with identifying the scattered lepton, but in practice the highest rapidity proton is always identified by the `DISDiffHadron` projection. In other diffractive analyses, the diffractive final state is identified by the presence of a rapidity gap amongst the hadrons. A `DISRapidityGap` projection exists to facilitate this.

### 3.5  Detector emulation

RIVET was initially developed to encode *unfolded* analyses, i.e. those for which the biases and inefficiencies introduced by detector material interactions and imperfect reconstruction algorithms have been corrected, making the experiment's published observables the best possible estimate of what happened at the fundamental interaction point, independent of any particular detector.

It remains our firm belief that unfolded measurements — while requiring a significant investment of time and effort to understand and invert the detector biases, and to minimise model-dependence in the derivation of such corrections — are the gold standard form in which to publish collider physics measurements. This is particularly the case when the fiducial analysis phase-space (i.e. the allowed kinematic configurations at truth-particle level) has been carefully designed to minimise extrapolation beyond what the detector could actually (if imperfectly) observe.

Not all collider physics analyses are appropriate for detector-unfolding, however. For example, stable unfolding relies on probabilistic algorithms to determine the probabilities of

event migration between truth-particle and reconstruction-level observable bins, and hence the MC populations used to derive these migration probabilities must be large enough to achieve statistical convergence. Some analysis phase-spaces, in particular BSM particle searches on the tails of distributions or on the edges of allowed kinematics, may be prohibitively difficult to simulate in the required numbers. Even if the MC samples can be made sufficiently large, the propagation of small numbers of observed events through the unfolding machinery can be fatally unstable, and also the low number of events present in the data means the ability to validate a large MC sample may be limited, unless appropriate control regions can be defined. Finally, the culture of BSM searches has historically been that speed is of the essence in the competition between experiments. Therefore, as unfolding — with its additional complexity and need for extensive cross-checking — does not intrinsically add exclusion power in e.g. the studies of simplified BSM models that LHC experiments use ubiquitously as phenomenological demonstrations of analysis impact, it has typically been neglected from search analyses. While this culture is necessarily changing in the high-statistics limit of LHC running, where an extra 6 months of data-taking does not automatically revolutionise the previous measurements, and in the realisation that simplified models are not always good proxies for full UV-complete BSM models [44, 84], it remains the case that with a few exceptions [85, 86], unfolding is currently rarely part of the vocabulary of collider BSM direct-search analyses.

It is in response to these factors that machinery for detector emulation has been added to RIVET— to permit the important class of reconstruction-level search analyses to be preserved for re-interpretation, albeit through an approximate detector model. The detailed implementation of this is reviewed in Ref. [87], along with comparisons to other initiatives [88–90] with similar intentions, but here we give a brief flavour of the features.

The key decision in implementing detector modelling was to use a "smearing + efficiency" approach rather than to attempt to model detector geometries, particle–material interactions, and thousands of lines of private reconstruction software within the RIVET package. This long-established approach distinguishes the RIVET detector modelling from that of the DELPHES fast-simulation code [91]. Specifically, we have chosen to implement detector effects as "wrapper" `SmearedParticles`, `SmearedJets`, and `SmearedMET` projections around the standard particle-level `ParticleFinder`, `JetFinder`, and `MissingMomentum` classes. These wrappers perform the dual tasks of modelling reconstruction efficiency losses, such that the wrapper may return a sampled subset of the particles (or jets) found by the contained truth-particle projection, and (except for the MET one) of "smearing" the 4-momenta of the truth-level objects to represent inaccuracies in kinematic reconstruction. Both the efficiency and smearing decisions are made using user-supplied functors (i.e. named functions, lambda functions, or stateful function objects) of the physics objects, respectively returning either a `bool` for efficiency filtering or a new `Particle`/`Jet` for smearing, with the sampled loss rate and smearing distributions dependent on the original object properties, most commonly their $|\eta|$ and $p_\mathrm{T}$. The advantage of this method, in addition to simplicity, is that it is fully customisable to measured efficiency and smearing effects in the specific phase-space of each analysis, and can be embedded directly in the analysis code, rather than relying on the universal correctness of a monolithic third-party detector simulation.

In addition to this machinery, a large number of standard efficiency and smearing parametrisations for ATLAS and CMS have been implemented, based on a mix of DELPHES configurations and experiment reconstruction performance papers [92–104]. These in turn are based on generic helper functions such as Gaussian $p_\mathrm{T}$ or mass smearers, $b$-tag efficiency/fake samplers, etc., which also act as a useful foundation on which users can build their own detector parametrisations. As with all RIVET tools, the implementation emphasises well-behaved default settings, and physics content over language noise.

## 3.6 BSM search analysis features: cut-flow monitoring

The object-filtering metafunctions and detector emulations described above constitute the main features that allow RIVET to now support the majority of LHC BSM search analyses. For example a search analysis can find leptons with e.g. `FinalState(Cuts::abspid == PID::MUON && Cuts::pT > 50*GeV && Cuts::abseta < 2.7)`, or `DressedLeptons(PromptFinalState(Cuts::abspid == PID::ELECTRON))`, then wrap them into a detector-biased form with a call to e.g. `SmearedParticles( elecs, ATLAS_RUN2_ELECTRON_EFF_TIGHT, ATLAS_RUN2_ELECTRON_SMEAR)`, which is then declared and applied like a normal particle-level projection. Jets found and smeared similarly can be isolated from the leptons using a few calls to functions like `discardIfAnyDeltaRLess(elecs, jets, 0.4)`. This makes for an expressive and powerful reconstruction-level analysis emulator, comparable to other tools on the market.

RIVET provides one more feature specifically targeted at BSM search implementation: a dedicated cut-flow monitoring tool. All analyses apply chains of event-selection cuts, but these are particularly crucial for re-implementers of search analyses because the cut-flow — i.e. the sequence of numbers or fractions of signal events passing each selection cut — is often the only published validation metric comparable in detail to the differential histograms of measurement analyses. Coding a cut-flow monitor by hand, e.g. via a histogram, is easy enough, but rather repetitive: one often has to write the cut logic once for the histogram fill, and then again for the actual event veto or signal-region iteration. The problem becomes particularly acute when, as is often the case, the analysis contains many parallel search regions, all of which have their own cut-flow. On top of all this, one needs to then code a print-out of the cut-flow stages, including raw event counts — possibly normalised to cross-section and luminosity, or to a fixed reference — as well as step-wise and cumulative efficiency fractions.

RIVET's `Cutflow` object exists to solve these problems. It acts like a fillable histogram, augmented with extra methods to attach human-readable labels to each cut stage, to track the current cut stage rather than make the user do so manually via a histogram's fill variable, to be able to fill multiple cut stages at once, and to return the result of a fill such that it can simultaneously update the cutflow and a signal-region counter, e.g. `signalregion[i]->fill(_cutflow->fillnext({pT1 > 50*GeV, aplanarity < 0.1}))`. Passing to a `stringstream` produces a nicely aligned text representation which can be printed with e.g. `MSG_INFO(_mycutflow)` in the analysis's `finalize()` stage, with the raw-count, step-wise efficiency, and cumulative efficiency measures all reported. `Cutflow::scale()` and `normalize()` methods are provided, the latter with a optional flag to determine which cut stage the normalization should be matched to. In addition, a map-like `Cutflows` wrapper is provided, for containing many named `Cutflow` objects, and potentially filling all of them with a single `fill()` call to the container; it is also able to write out its contained cut-flows in the standard form. These small but significant features make debugging and validating a search analysis a more pleasant experience.

## 3.7 Analysis options

From the beginning it was assumed that an analysis in RIVET is completely specified by the corresponding `Analysis` class and could only be run in one way. However, especially with the introduction of heavy ion analyses, it became clear that some analysis has to be treated differently depending on which event generator it is used for. In particular for the centrality concept used in heavy ion analyses, there are different ways of handling them (see Section 3.3). Similar issues arise for analyses based on a single paper, but in which cross-sections are measured for e.g. muons and electrons separately and also combined, or for different definitions of heavy-flavour-tagged jets. In such cases it is efficient to be able to specify different running

modes for a single analysis.

For this reason RIVET now includes an option machinery. For any analysis name it is possible to specify one or more options that can take on different values. This is encoded by supplying suffixes to the analysis names on the form

```
rivet -a AnalysisName:Opt1=val1:Opt2=val2
```

In the analysis class it is then possible to retrieve the specified value of a particular option with the function `string Analysis::getOption(string)`.

It is possible to specify the options and values that are allowed for a given analysis in the `.info` file, but it is also possible communicate other options to an analysis.

Note that it is possible to use several versions of the same analysis in the same RIVET run as long as the versions have different options.

Handlers are provided in the Python interface to extract or strip the options from histogram paths.

## 3.8 Dependency evolution

In addition to explicit RIVET framework developments, version 3 supports new versions of dependencies, both physics libraries and the Python runtime.

### HepMC3

The default configuration of RIVET currently assumes that HepMC version 2 is used for the event files to be read by RIVET. In the future this will be changed to compiling RIVET with HepMC3. Already now it is possible to try out HepMC3 with rivet by providing the flag `--with-hepmc3=/path/to/installation` to the `configure` script.

Note that when compiled with HepMC3, Rivet can still read HepMC2 files. In fact RIVET will then automatically detect the format the given event file.

### Python 3

While the RIVET core library and the analysis implementations are written in C++, there is a full featured Python wrapper around them built using the Cython system. In fact, the `rivet` executable is written in Python and uses this interface.

Cython is not a requirement to install the RIVET Python modules if building from a release tarball, but is necessary when building from the git repository directly.

Recent RIVET versions are compatible both with Python 2.7 and Python 3. If multiple Python versions are available in a system, you can choose which one to use for the RIVET installation by prefixing the configure command with e.g. `PYTHON=/usr/bin/python2 ./configure ...`. It is also possible to install modules for both versions in the same installation location by running the appropriate `PYTHON=<python> ./configure ...; make install` twice.

## 4 User guide

Here we provide a short user guide to help with getting RIVET 3 up and running both standard built-in analyses, and your first analysis routines. Full information, including full code API documentation, can be found on the RIVET website at https://rivet.hepforge.org/, and the YODA one at https://yoda.hepforge.org/.

## 4.1 Installation

Getting started RIVET is most easily done using the DOCKER images installed via `docker pull hepstore/rivet`, then entering an interactive environment with RIVET available with e.g. `docker run -it hepstore/rivet`. Many useful variants on these commands can be made, for example using an image with an MC generator also included, such as `hepstore/rivet-pythia`, or using the `-v` flag to `docker run` to mount host-system directories inside the image so external event files can be easily read in, and histogram files written back out. Alternatively, a native installation can be made easily on most *nix systems by downloading a "bootstrap script" which will build RIVET and all its dependencies — full instructions for this are provided on the website. After installation and set-up of the RIVET environment, it can be accessed at the command-line using the command `rivet`: for example, try getting the list of supported commands and options using `rivet --help`.

## 4.2 Analysis metadata inspection

The first point of call in using RIVET is finding which analyses are of interest. There are over 900 analyses in the current release, as you can verify by running `rivet --list-analyses` and counting the number of resulting lines, and so being able to find the ones you want requires some searching. A full list of standard analysis routines, with information about the collider, energy, luminosity, date, process type, an abstract-like description, bibliographic data, and a syntax-highlighted copy of the analysis code, can be found on the RIVET website. The information used to build these web pages is also accessible from the command-line, with for example `rivet --list-analyses ATLAS_2018` being usable to print a one-line description of each routine whose name contains the pattern "ATLAS_2018" (corresponding to ATLAS experiment papers published in 2018), and `rivet --show-analyses ATLAS_2018` printing out the fully detailed metadata entries for each analysis matching the pattern. You will notice that the majority of analyses have a standard name of this kind: ⟨*expt*⟩_⟨*year*⟩_⟨*Innnnnnnn*⟩, where the last portion is an ID code corresponding to the analyses key in the Inspire publication database [105].

## 4.3 First RIVET runs

Now to run analyses on simulated events, using pre-written analyses. Let's say we want to analyse some top-quark pair ($t\bar{t}$) events at $\sqrt{s} = 13$ TeV, using routines that both compare to real data, and generically characterise the jet and partonic top distributions in the events. Using the metadata search system on the command-line we find the `CMS_2018_I1662081` data analysis, and also the `MC_JETS` and `MC_TTBAR` routines which have no experimental analysis counterpart. These analyses are located by the RIVET analysis loader system, which by default looks in the RIVET library install path under `$prefix/lib/Rivet/`. If the `$RIVET_ANALYSIS_PATH` environment variable is set, or search paths are specified via the RIVET library API, these are used by preference with fallback to the default unless the path variable ends with a double-colon, `::`.

If you have a set of 10k–1M $t\bar{t}$ events in HepMC format, then running is trivial — just tell `rivet` which event file(s) to read from, and which analyses to run, e.g. `rivet -a CMS_2018_I1662081 -a MC_JETS,MC_TTBAR events.hepmc`. Analyses can be specified both by multiple instances of the `-a` option flag, or by comma-separating analysis names in the arguments to a single `-a`, as shown. The event file may be either uncompressed, or gzipped, but must be supported by the HepMC library.

More commonly, especially for large MC event samples, we generate the parton-showered events "on the fly", and pass them directly to RIVET. This can be done most efficiently by using

the RIVET C++ library API to hand HepMC objects in memory between the event generator and RIVET, and so requires either built-in RIVET support in the generator (as for SHERPA [106] and HERWIG [3]), or for the user to write a C++ program that uses both libraries (as is the case with PYTHIA 8). A slower, but completely generator-agnostic, way is to write out a temporary event file and read it in to RIVET: for this, Unix systems have a very useful feature in the form of a "FIFO file" — a file-like object for inter-process communication. To run RIVET this way, first make a FIFO with e.g. `mkfifo myfifo.hepmc`, then run the generator *in the background* (or in a separate terminal on the same system) with instructions to write HepMC-format events out to the FIFO: `some-generator --some-config=ttbar --out=myfifo.hepmc`. Finally, run `rivet` as before: the generator writing and RIVET reading will control each other such that events are passed between them and the "file" never gets any bigger than a few tens of kilobytes.

RIVET will happily chug through the provided events, updating an event counter on the terminal and periodically writing out a `.yoda` file containing output histograms and counters from the `analyze()` and `finalize()` stages of the RIVET analyses' processing. Using the tools described in the following section, you can inspect and plot these intermediate files should you wish. If you find that you have acquired sufficient statistics, and don't need the rest of the generator file or run, you can perform the usual `Ctrl-C` intervention to kill the `rivet` process, which will exit gracefully after `finalize`ing the remainder of `analyze`d events.

## 4.4 Plotting and manipulating results

The usual next step is to plot the results. The final `.yoda` file written by the `rivet` run (named `Rivet.yoda` by default) is the principle input to this plotting, optionally along with equivalent files from other MC runs.

If multiple MC runs — either for separate MC processes or to split a large single homogeneous-process run into smaller chunks — need to be combined into a single `.yoda` file for this purpose, the `rivet-merge` script can be used to read in these independent contributions and re-run the analyses' `finalize()` methods to give a final, statistically exact combined `.yoda` file, as described in Section 3.2.4. Cross-section and number-of-events scalings will be automatically calculated from information stored in the input files. Should any manual scaling be needed in addition, the `yodascale` script or a custom manipulation using the YODA Python API are also possible.

The usual approach to plotting is to run the `rivet-mkhtml` script. This is a wrapper around the lower-level `rivet-cmphistos` and `make-plots` scripts, which respectively group sets of histograms by analysis, and render them to PDF format, with additional generation of HTML code and thumbnail images so the output can be conveniently viewed via a Web browser. Reference data will be automatically loaded from the same location as the compiled analysis library (or more generally from the `$RIVET_DATA_PATH` path list).

## 4.5 Basic analysis writing

The writing of analyses is as potentially multi-faceted as writing any computer program, and hence cannot be covered here in comprehensive detail. The best way to learn, as ever, is by doing and by consulting existing analysis routines with similar ambitions to your own. But compared to a completely general program, RIVET routines are constrained by the three-step `init`/`analyze`/`finalize` event-processing structure, and by their necessary inheritance from the `Rivet::Analysis` type: here we will survey the main features of each step.

**Raw materials**

Our first step is to generate the file templates into which we will insert analysis data and logic. The starting point should always be the `rivet-mkanalysis` script, run like `rivet-mkanalysis EXPT_2019_I123456` where the analysis name follows the three-part structure described earlier. Particularly critical is the third part, encoding the Inspire database key with which the script can automatically extract reference data and publication metadata.

Running the script in this way generally results in four template files: a `.cc` file containing a generic template for your analysis code, with "boilerplate" features pre-filled; a `.info` metadata file in YAML format, used to generate documentation and constrain applicability; a `.plot` file used to apply plot styling directives to sets of output histograms; and a `.yoda` reference data file in the YODA format, downloaded if possible from the HepData database. The only essential file for private running is (of course) the `.cc` in which the analysis logic will be written, but any analysis submitted to RIVET for official inclusion in the repository must also complete the other files. In particular it is critically important that the `.yoda` file be compatible with HepData's YODA output, so updates can be synchronised with subsequent RIVET releases.

**Projections**

Projections are the engines of RIVET: they are calculators of observables, encapsulating various nuances in the definitions and efficiency insights for calculation, as well as benefiting from the automatic caching of their results. The most important projections are those which inherit from the `ParticleFinder` interface: these include the `FinalState` projection which extracts and returns subsets of stable final-state particles; its specialised children like `PromptFinalState` which excludes final-state particles from hadron decays, and `VisibleFinalState` which only returns particles that would interact with a detector; composite final-state particle finders like `DressedLeptons` which sums prompt photons in cones around charged leptons; decayed particle finders `UnstableParticles` and `TauFinder`; and pseudoparticle finders like `WFinder` and `ZFinder` which reconstruct leptonic composites at EW scale by experiment-like fiducial definitions. Other important projections are `FastJets`, the main implementation of the `JetFinder` interface, and `MissingMomentum` for a particle-level definition of missing $E_T$. Using these projections affords the analysis author quick and robust definitions of physics objects and quantities from which the rest of the analysis logic can be applied.

The caching machinery around projections means that they must be used in a slightly non-intuitive way: they are *declared* in the `init()` method of an analysis, and then retrieved and *applied* in the `analyze()` step. The declaration involves first constructing and configuring a local object in the `init()` method, e.g. `FinalState fs(Cuts::pT > 10*GeV);` and then assigning a string name to it, e.g. `declare(fs, "MyFS");`. The string name must be unique within this analysis, but different analyses are free to use the same names. Once declared, the projection object has been cloned into the RIVET core, and the local copy will be automatically deleted once the `init()` method closes. Then in `analyze()`, the projection's computation is performed by referencing the registered name, e.g. `FinalState& pf = apply<FinalState>(event, "MyFS");`. In fact it is common to bypass the projection itself in the application, going straight to the result of one of its methods, e.g. `const Particles ps = apply<FinalState>.particles();`.

**Histograms and counters**

Statistics objects in RIVET must, like projections, be managed by the system core: this is to enable the automatic handling of event-weight vectors, including details such as fractional fills and counter-event groups (cf. Section 3.2), as well as run-merging and re-entrant calls

to the `finalize()` function. For efficiency, convenience and flexibility in how they are handled by user code, references to YODA histograms, profile histograms, and weight counters are stored as `Rivet::Histo1DPtr`, `Rivet::Profile1DPtr`, `Rivet::CounterPtr`, etc. member variables within the analysis class. The actual histogram configuration is performed on these using a set of overloaded `Analysis::book(hptr, ...)` methods, where `hptr` is any of these `Rivet::*Ptr` objects. For binned histograms, the remaining arguments can be a ROOT-style (*hname*, $N_{bins}$, $x_{min}$, $x_{max}$) tuple, a (*hname*, [$x_{edges}$]) pair, or a single name string corresponding to the reference data histogram whose binning should be used by its RIVET equivalent. This latter form also has an integer-triplet shorthand, expanding to the `daa-xbb-ycc` dataset/axes format output by HepData. Counters and `Scatter*D` objects, which do not have a variable binning, have simpler `book()` method overloads. Note that analyses which run in several modes, e.g. making the same kinds of observable histograms for event runs at two different $\sqrt{s}$ energies, so not need different histogram pointer variables for each mode — simply pass different additional arguments to the `book()` methods depending on the context of, for example, a call to `Analysis::sqrtS()/GeV`.

Within the analysis, the `Ptr` objects are used in a straightforward fashion, most notably calls like e.g. `_h_myhist->fill(x)`. Users of RIVET v2 will note that the event weight is no longer part of this function signature. In fact, attempts to call `Event::weight()` will now be spurned, returning `1.0` and a warning message. This is because in RIVET v3, there is no single event weight and it is the job of the RIVET *system* rather than the user to handle and combine weight vectors correctly. Weighted fills are still allowed, but for weights than the event ones. Behind the scenes, the `Ptr` objects are multiplexed on to arrays of fundamental YODA objects, with RIVET secretly looping over weight arrays or running methods once for each weight, but users can and should act happily oblivious to these sleights of hand.

The biggest such trick occurs between the `analyze()` and `finalize()` methods, when the active histograms (or histogram *sets*) are persisted as "raw" versions, to allow for pre-finalize run combination. Within finalize, each weight stream is treated independently, but again users need not worry about these details. The most common finalising operations are calls to `normalize()` or to `scale()`, which respectively fix the area of a histogram to a fixed number (perhaps proportional to the process cross-section obtained via e.g. `crossSection()/femtobarn`) or scale it by a factor. In analyses where not every event results in a histogram fill, the latter approach is usually what is wanted if measuring absolute cross-sections, e.g. `scale(_h_myhist, crossSection()/picobarn/sumW())`: this division of the histogram's accumulated sum of cut-passing event weights by the all-event process cross-section `sumW()` encodes an acceptance factor for each weight-stream into the final histogram normalization. `Counter` objects may also be useful for this acceptance tracking. For convenience, the `normalize()` and `scale()` methods also accept histogram containers, either explicit or via an initialisation list cf. `normalize({_h_pt, _h_eta, _h_mass})`. Both the raw and finalized statistics objects are automatically written out, both periodically and at the end of the run, exactly as for "official" RIVET routines.

**Analysis logic and tools**

Having established the analysis class structures, and the core machinery of declared projections and booked histograms, the "only" remaining part is the logic of your specific analysis as acted out with these participants. Here there is relatively little to say: the principle logic and control flow tools are simply the syntax of procedural/object-oriented C++: `for` and `while` loops, `if ... else` statements, boolean expressions, ternary `x ? y : z` syntax, etc.

Any feature of the STL can also be used, with standard types like `std::vector` and `std::map` already imported into the `Rivet` namespace and more clearly referred to simply as `vector` and `map`. For convenience, vectors of some standard types are given convenience

aliases, e.g. `Particles`, `Jets`, `doubles`, `strings`, and `ints`. The first two in this list are RIVET-specific types which it is worth becoming familiar with in some detail, as they support not just kinematic operations like `pT()`, `eta()`, `absrap()`, etc. (as described earlier in this document), but also ways to interrogate their composite nature, decay ancestry, connection to `HepMC::GenParticlePtr` objects, etc. `Jet` objects additionally allow for *b*- and *c*-hadron truth-flavour labelling using a robust and now-standard ghost-association definition.

Note that all unitful kinematic quantities such as `ParticleBase::pT()`, `mT()`, etc. (as well as the return value of `Analysis::crossSection()`) are returned as `doubles`, but should be treated in this form as having an undefined "default RIVET unit": they are not safe to be passed to a YODA histogram without an explicit unit declaration. The definition used for this is that multiplying by a unit constant converts the numerical value from that unit to the RIVET default unit, and dividing will convert back to a number representing how many of the dividing unit the RIVET internal value corresponded to. Hence the RHS term in `Cut::pT > 10*GeV` converts from ten GeV units to RIVET's internal scheme, and `hist->fill(p.pT()/GeV)` is a conversion to the unitless number to be plotted on a GeV axis.

Several other additional features have already been described, such as the filtering `select()`, `reject()`, `iselect()`, etc. functions, and the many functors to be used with them. Many examples of best-practice usage may be found in the code and in dedicated tutorials available from the RIVET Web page.

## 4.6 Building and running analysis codes

Having written the C++ analysis code, it must be built into a compiled "plugin" library that can be loaded by the RIVET system. This is done at runtime without needing the core RIVET framework to be recompiled, using the C `dlopen()` dynamic loader. Since the core library and plugin must be compatible at compiled binary code level, the C++ compiler used for plugin building must see exactly the same headers from RIVET dependency packages like HepMC and FastJet: this leads to a complex compiler command line that could easily be a source of mysterious technical crashes or strange behaviour, and hence a convenience wrapper script, `rivet-build`, is provided to encode all the necessary compiler incantations. It is run on any number of analysis source files like `rivet-build MYANA1.cc MYANA2.cc ... -lextralib -extra_option` and produces a `dlopen`able shared library named `RivetAnalysis.so`.

Obviously it would be awkward if all analysis plugin libraries had to have the same filename, and so a custom output name can be given as an option first argument in the form `Rivet*.so`. When running RIVET, the library will search the `$RIVET_ANALYSIS_PATH` variable and installation prefix fallback (as previously described) for `.so` libraries matching this pattern, from which all the contained analyses will register themselves with the RIVET core. As a convenient shorthand for the path variable setting, the `rivet` script (and related tools like `rivet-mkhtml` and `rivet-merge`) can take an optional `--pwd` flag, equivalent to prepending `$PWD` to the analysis and data search paths.

## 4.7 Contributing analysis routines

We encourage all users to contribute validated analyses to the official RIVET collection: this helps the whole particle physics community, and your efforts will be accredited through public analysis authorship declarations. After submission, the RIVET core authors will be responsible for maintaining the analysis code compatibility with any future changes in the core API.

In addition to the `.cc` code file, the metadata `.info` file, plot styling `.plot` file, and if appropriate `.yoda` reference data files must be provided, along with information (ideally including plots from `rivet-mkhtml`) illustrating the pre-submission validation procedure performed by the analysis author. The info file must include a `ReleaseTests` entry indicating

how a short analysis-validation behavioural regression run should be performed, using the example 1000-event HepMC event files located at http://rivetval.web.cern.ch/rivetval/. If no existing event file is suitable for the analysis, a new `.hepmc.gz` analysis file should be supplied along with the analysis code and metadata upload.

In the past, contribution of analyses and this supporting validation information has been done through a mix of email submissions to the RIVET developer mailing list and (for official experiment representatives) upload to a special "contrib" area on the RIVET website. Since version 3.0.2, a more formal procedure is in operation, whereby new analyses are to be contributed via merge requests on the https://www.gitlab.com/hepcedar/rivet code repository. Validation plots and similar information, and new HepMC event samples if appropriate, should be contributed to the RIVET core team separately from the repository merge request.

Our thanks in advance for contributing to this important community analysis preservation resource!

## 5 Conclusions and future plans

Over the last decade, RIVET has become established in the ecosystem of particle physics analysis preservation, primarily for, but not limited to, the LHC experiments. Its position in this world is an intermediate one, using more complex and complete particle-level final states than in partonic matrix-element analysis tools, while less fully detailed (and computationally expensive) than forensic preservations of experiment simulation and analysis frameworks. This mixture of detail and efficiency has led to many uses in studies from MC generator model and tune optimisation, to limit-setting studies on explicit and effective Beyond Standard Model theories.

In this review we have noted how the core set of "RIVETable" analyses, formalised as fiducial phase-space definitions, have become part of standard operating procedure for detector-corrected measurements, while the remit of RIVET has expanded to include more complex multi-pass observables as used in heavy ion physics, and approximations of reconstruction-level quantities, particularly new-physics searches. RIVET has also evolved to take advantage of new developments in the precision and control over calculational systematics from Monte Carlo event generators, in the form of an unprecedentedly transparent handling system for event weights and counter-event groups. The full realisation of these features, and extensions to yet more areas of particle physics such as astroparticle and neutrino physics, is a challenge for the next decade of RIVET and of particle-physics data- and analysis preservation.

## Acknowledgements

We thank all contributors of analysis codes, bug reports, and code patches and improvement suggestions for their efforts to make RIVET the best possible tool for analysis preservation and interpretation. Our thanks also to the MC generator author community for their support of the RIVET project, and all those who hosted the meetings and workshops that informed and accelerated its development.

**Author contributions** Project leadership, and evolution of fiducial analysis concepts: AB, JMB. Core machinery, cuts system, and projection design and implementation: AB, LC, LL, DG, CG, HS, FS. Multi-weight, re-entrant finalize, and histogramming system: AB, LC, DG, CG, CP, LL. Design and development of heavy ion analysis features: CB, CHC, AB, JFG-O, PK, JK, LL. Refinement of *ep* physics tools and definitions: AB, JMB, LL. Detector emulation system and

search tools cf. filtering functors: AB. Validation and low-$Q$ / hadronic decay physics coverage: PR

**Funding information**   This work was supported in part by the European Union as part of the Marie Sklodowska-Curie Innovative Training Network MCnetITN3 (grant agreement no. 722104). AB thanks The Royal Society for University Research Fellowship grant UF160548. AB and CP thank the University of Glasgow for postdoctoral funding through the Leadership Fellow scheme. LL was supported in part by Swedish Research Council, contract number 2016-03291. FS was supported by the German Research Foundation under grant No. SI 2009/1-1. Many thanks to Xavier Janssen, Markus Seidel, Alex Grecu, and Antonin Maire for (in addition to members of the author list) acting as LHC experiment contacts.

# A   Handling groups of completely correlated events

The problem addressed here is how to process histogram fills in an event group with fully correlated events, in a way such that the it is treated as one event fill and still have the correct error propagation in terms of the SoSW. It is also essential to make sure that large cancellations remain cancelled across bin edges. The solution is to introduce *bin smearing* and the concept of a *fractional fill*.

As of YODA version 1.6, the histograms have an extra argument to their fill functions, encoding the concept of a fractional fill. This means that instead of having one fill with weight $w$, we divide it up in $n$ fills with weights $f_i w$ where $\sum f_i = 1$. For the single fill we will add $w$ to the SoW and $w^2$ to the SoSW. The SoW is no problem, we simply add up the fractional fills: $\sum f_i w = w$, but the naive approach of doing $n$ fractional fills would give a contribution $\sum (f_i w)^2 \neq w^2$ to the SoSW. The solution is obviously that a fractional fill should instead contribute $f_i w^2$ to the SoSW, giving the result $\sum f_i w^2 = w^2$ for the $n$ fills, which is what we want.

Now we look at the case where we have $N$ sub-events in an event group which are fully correlated as in a NLO calculation with a real correction event and a number of counter-events. Let's assume that we are measuring jet transverse momentum and we have one fill per jet for $M$ jets. We have one weight per sub-event, $w_i$, and we apply smearing such that each jet, $j$, is filled with a fraction $\epsilon_{ji}$ in one bin and $1 - \epsilon_{ji}$ in an adjacent bin. Clearly, if these bins were joined, we would like to have

$$\text{SoW} = \sum_{j,i} w_i \quad \text{and} \quad \text{SoSW} = \sum_j \left( \sum_i w_i \right)^2. \tag{14}$$

As before SoW is no problem, we simply fill

$$\sum_j \sum_i \epsilon_{ji} w_i + \sum_j \sum_i (1 - \epsilon_{ji}) w_i = \sum_{j,i} w_i. \tag{15}$$

For the SoSW we clearly cannot fill

$$\sum_j \sum_i \epsilon_{ji} w_i^2 + \sum_j \sum_i (1 - \epsilon_{ji}) w_i^2, \tag{16}$$

since we need to combine all the fills in the event group before filling. Nor can we do

$$\sum_j \left( \sum_i \epsilon_{ji} w_i \right)^2 + \sum_j \left( \sum_i (1 - \epsilon_{ji}) w_i \right)^2, \tag{17}$$

since we need to take into account the fractional fills. The trick is now to realise that for a given jet, $j$, the NLO calculation requires that the fill values are close to each other so also the $\epsilon_{ji}$ are close to each other. We can therefore replace them with the average $\bar{\epsilon}_j = \sum_i \epsilon_{ji}/N$ and fill SoSW with

$$\sum_j \bar{\epsilon}_j \left(\sum_i w_i\right)^2 + \sum_j (1 - \bar{\epsilon}_j) \left(\sum_i w_i\right)^2 = \sum_j \left(\sum_i w_i\right)^2, \tag{18}$$

which gives the correct SoSW. However, the averaging of $\epsilon_{ij}$ means that we are assuming that the NLO calculation is stable, and the errors will be calculated as if it was stable. We would like to have a procedure where the errors becomes large if the NLO calculation is unstable. Therefore we need a procedure which takes into account that the $\epsilon_{ij}$ are not exactly the same for a given $j$, while it still gives the result above if they are.

For two sub-events, the procedure should be such that if $\epsilon_{1j} < \epsilon_{2j}$ we should have two fractional fills, one filling $w_1 + w_2$ with fraction $\epsilon_{1j}$, and one filling only $w_1$ with fraction $\epsilon_{2j} - \epsilon_{1j}$ (and the corresponding fills in the neighbouring bin).

For more sub-events, it becomes a bit more complicated, and even more complicated if it is not obvious which jet in one sub-event corresponds to which in another.

RIVET 3 implements a procedure defining a rectangular window of width $\delta_{ij}$ around each fill point, $x_{ij}$. This width should be smaller than the width of the corresponding histogram bin, and as $x_{ij} \to x_{ik}$ we should have a smooth transition $\delta_{ij} \to \delta_{ik}$. As an example we could use a weighted average of the width of the bin corresponding to $x_{ij}$ and the width of the closest neighbouring bin. So if we have bin edges $b_k$ and the bin corresponding to $x_{ij}$ is between $b_k$ and $b_{k-1}$ centred around $c_k = (b_k + b_{k-1})/2$ we would take

$$\delta_{ij} = \begin{cases} x_{ij} > c_k : \varepsilon \left( 2(b_k - x_{ij}) + \frac{x_{ij} - c_k}{b_k - b_{k-1}}(b_{k+1} - b_{k-1}) \right) \\ x_{ij} < c_k : \varepsilon \left( 2(x_{ij} - b_{k-1}) + \frac{c_k - x_{ij}}{b_k - b_{k-1}}(b_k - b_{k-2}) \right) \end{cases}, \tag{19}$$

with $\varepsilon < 1$.

The procedure in RIVET 3 is therefore the following:

- Collect all fills, $x_{ij}$ in all $N$ sub-events with weight $w_i$.

- Construct the smearing windows $\delta_{ij}$.

- Set the fill fraction to $f_{ij} = 1/N$ (since we want each jet to sum up to one fill).

- Construct all possible sub-windows from the edges, $x_{ij} \pm \delta_{ij}/2$, of all windows.

- For each sub-window, $l$, with width $\delta_l$, sum up all fills which are overlapping (they will either completely overlap or not at all) and sum up

$$w_l = \sum_{\delta_l \in \delta_{ij}} w_i, \tag{20}$$

$$f_l = \sum_{\delta_l \in \delta_{ij}} f_{ij} \delta_l / \delta_{ij} \tag{21}$$

and fill the histogram bin corresponding to the midpoint of the sub-window with weight $w_l$ and fraction $f_l$.

Figure 2 provides a pictorial illustration of how the sub-windows are constructed for a one-dimensional histogram. This procedure is easily extended to two-dimensional histograms, which will done in a future release.

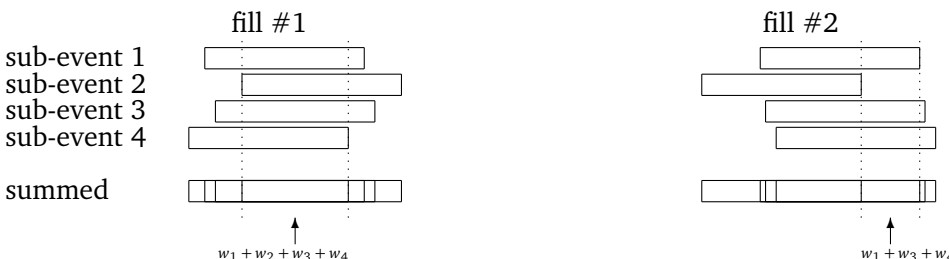

Figure 2: A pictorial view of how the sub-windows are constructed for a one dimensional histogram.

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
