# Peer review of "Robust Independent Validation of Experiment and Theory: Rivet version 3"

_SciPost Physics, doi:SciPost Phys. 8, 026 (2020)_

## Round 2 · Referee Report · Anonymous (Referee 1) · 2020-1-11

Strengths

(1) Rivet is a freely available software tool that aids preservation and reinterpretation of data from high energy physics experiments.

(2) The paper acts as a manual to enable new users to understand and easily use the code. This is augmented by online resources and example routines.

(3) Version 3 of Rivet in this paper introduces a number of new features, such as the ability to use multiple event weights and new options for Heavy Ion collisions. These new features are a major step forward for the community. The implementation of many of the new features in Rivet is subtle, with the majority of the code implemented behind the scenes, such that the user is not concerned with technical aspects but can easily access the results.

(4) The paper is very well written and easy to understand

Weaknesses

None

Report

This paper documents version 3 of the Rivet software framework. Rivet is an invaluable tool used by the High Energy Physics community in terms of data preservation and reinterpretation. Version 3 introduces a number of features long sought after by the HEP community. The paper is well written and should be published immediately.

Requested changes

None, though the authors may want to consider that the term "two finger salute" (p23) has various meanings, one of which is perhaps offensive in some countries.

  • validity: top
  • significance: high
  • originality: good
  • clarity: top
  • formatting: excellent
  • grammar: excellent

Author:  Andy Buckley  on 2020-02-06  [id 729]

(in reply to Report 1 on 2020-01-11)

Thanks for the very positive review. The two-finger salute has been rephrased in the new version, just submitted to arXiv.

Andy, for the authors

---

## Round 2 · Referee Report · Anonymous (Referee 2) · 2020-1-17

Strengths

1 - The paper compiles important information about recent developments, updates and additions to the Rivet analysis package, giving an insight into design decisions

2- The paper provides a very useful documentation of available methods and features

3- It compiles relevant references for further reading on the foundations of several of the new features

Weaknesses

1- In particular the introduction lacks a guide for the reader what to expect about the content of the paper and why certain aspects needed to be addressed

2- The quality of the text and descriptions varies between the different sections

3- Some relevant references are missing

Report

Rivet is a well established and widely used analysis framework. It allows for the validation of Monte Carlo event generators against actual experimental data. Furthermore, it provides means for the experimental collaborations to preserve their analyses and to make them accessible for use by the whole particle physics community.

The paper at hands compiles information about the latest major version of Rivet, v3, 10 years after the document describing the general framework, published 2013 in Comput.Phys.Commun. The developments that happened since are significant and certainly deserve a new publication. The focus of the paper is thereby on the major feature additions and design changes that have been developed and are mostly not (yet) published elsewhere. As such the paper supersedes the older version and largely replaces it.

However, the prior to publication I would like the authors to address a few points of criticism, related to the actual text and its structure:

I am afraid that for a reader not very familiar with Rivet the access to the paper is rather hermetic and certainly can be improved. The authors should reconsider which readers they want to address and how they guide them through the text. See my comment below.

Furthermore, the level of care given to the actual text varies between the sections. The authors should carefully re-iterate the text and correct several instances of wrong grammar, or missing words. See below for an incomplete list of corrections.

Furthermore, the way the authors approach the reader changes from a passive tone to a quite casual 'you' in particular in Sec. 4.2 & 4.3. I would propose to adapt these parts to the rest of the paper.

I hope that these suggestion can further improve the quality of the paper such that it can published in SciPost.

Requested changes

1- The introduction is kept extremely short, providing no summary of the things to be reported. The authors should more early on describe the motivation for the recent developments and the corresponding extensions/modifications to the Rivet system

As a minimum the authors should extend Sec. 1 by a guide to the reader, explaining the structure of the paper.

Along that line, Sec. 2 could be structured a little better as currently it mixes general information about the Rivet logic with rather detailed implementation or code specifics.

2- The authors are missing a few relevant references that I would ask them to add:

The HepMC package (and obviously the HepMC3 that appeared after this paper) is note referenced though it is clearly central for Rivet.

Is there any reference that could be given for YODA?

The event generators Herwig and Sherpa are quoted on p23 but no references are given.

On page 9 the authors report on the appearance of counter-events in NLO calculations, as a motivation for the new event weight treatment. They should reference to some corresponding publication, for example the Catani-Seymour subtraction method. Furthermore, I believe that the statement they make about the exchange of a virtual parton exchange representing the counter-events is misleading. These are in fact events that have the real-emission kinematics, resembling the singularity structure of possible soft/collinear emissions.

3- As mentioned before, I would like the authors to reconsider the way to address the reader in Secs. 4.2 & 4.3.

4- lastly, here is a list of grammar mistakes and wording errors I have spotted:

p9, para1 " ... generation ->of<- extremely large ... " p9, para5 "This means that ->it<- is not ..." p10, Sec.3.2.2, para1 "fill is given ->by<- the sum ..."

p13, Eq. (7) ",." -> "."

p14, last line "contains heavy ion beams" -> "containing heavy ion beams"

p15, Sec. 3.3.1 " ... in one ->of<- three ways ... "

p16, first line "... with respect ->to<- the event ... " p16 " ... the Generic Framework express->es<-"

p17, Sec. 3.4 "... community worked toward ... " -> "community work towards"

p17, Sec. 3.4.1 "... with options to selection ... " -> " ... with options to select ..."

... plus several more ... please re-iterate the whole manuscript carefully

  • validity: high
  • significance: high
  • originality: high
  • clarity: good
  • formatting: good
  • grammar: good

Author:  Andy Buckley  on 2020-02-06  [id 730]

(in reply to Report 2 on 2020-01-17)
Category:
answer to question

Thanks for the detailed feedback: we have added references and fixed language errors as specified, and have given the document an overhaul as suggested. A document-structure section has been added to the end of Section 1; and Section 2 is now more structured.

We have chosen to retain the more informal style in the tutorial, since the aim of that section is to communicate effectively and pedagogically, stepping a user through the system: using more passive and formal language would not obviously assist in that endeavour.

Best wishes,
Andy, for the authors

---

## Round 2 · Referee Report · Tilman Plehn (Referee 3) · 2020-1-22

Report

This is a nice paper describing a major update of a standard tool, including the right mix of physics, techniques, examples, and instructions.

Requested changes

1- Since we share an interest in unfolding, I have some questions concerning Sec 3.5. What remains a little unclear is where unfolding is meant to start for hadronic objects: hadrons or jets? And I agree that it does not make sense to unfold individual events, but that might not be obvious to others (we also did not write this in our paper and will add it) 2- Is it fair to claim that ATLAS and CMS do not systematically unfold? In top physics it's quite standard, the biggest problem is probably Higgs physics.

A few comments on the text: 3- why do the authors introduce quotes all over the place? Is that standard English? 4- what is an event-shape tensor? Googling it turns up mostly Rivet material. 5- I do not understand the discussion on p.12, sorry. What's the point? 6- why is there the extra paragraph about HERA? Is that not a quite useless experiment of purely historic interest? 7- on p.18 the `in vacuo' sounds slightly pompous, and it's not an established technical term. 8- the 6-months comment is funny, but then it might divide the audience along the lines of sense of humor. 9- please mention that the smearing and efficiency approach is old, is that not what ATLFAST and all its contemporaries did?

And I did not check the installation guide myself, hope it works as described...

  • validity: top
  • significance: top
  • originality: good
  • clarity: top
  • formatting: perfect
  • grammar: -

Author:  Andy Buckley  on 2020-02-06  [id 728]

(in reply to Report 3 by Tilman Plehn on 2020-01-22)
Category:
answer to question

Thanks for the comments, Tilman: we've taken these into account in a new arXiv submission made just now. Some responses to each of your points:

  1. This is a question about fiducial definitions, which is obviously of interest to Rivet but a wider issue in general: we have to work with what the experiments choose, as far as possible, and to feed back constructive criticism if experiment definitions can't be robustly implemented in Rivet. We've clarified the "conceptual barrier" for well-defined unfolding objects in the text now, as being defined by hadronisation, but the stage which is appropriate for a measurement is defined by a mix of insight provided on the fundamental physics, vs uncertainty inflation through detector and model dependence.

  2. The comment here is specific to direct BSM searches. SM, Top, and to some extent Higgs do unfold -- this corresponds to the majority of analyses in Rivet, but direct searches at reco level is a new thing for us.

  3. Yes, pretty standard English: they are used the first time that a new or potentially unfamiliar nomenclature (for this context) is introduced. Not for the benefit of experts, who will already know the jargon specific to this area.

  4. Rivet material and the Pythia manual which defines the sphericity tensors. Now made explicit and with a reference to the original Bjorken paper on sphericity.

  5. The point of this is fully consistent combination of weighted data objects, up to second-order moments, from different runs of Rivet either with the same or different process types. This underlies the re-entrant histogramming system and the systematics weight-vector handling. Appendix A extends this to handling of NLO counter-events, where groups of correlated events must have their weights aggregated before filling.

  6. Ex-HERA members object! No, it's still relevant ;-) But the itemizing of HERA analyses has been reduced to an equivalent of the HI analyses in the previous section.

  7. Established in other areas, and I rather like the analogy as a characterisation of idealised MC events. But no point in insisting if it annoys people, so I've removed it.

  8. Wasn't even meant to be a big joke! We're happy with this characterisation: can't please everyone.

  9. Now mentioned.

Thanks! Andy for the authors

---

## Editorial Decision

published